# Determination of Osimertinib, Aumolertinib, and Furmonertinib in Human Plasma for Therapeutic Drug Monitoring by UPLC-MS/MS

**DOI:** 10.3390/molecules27144474

**Published:** 2022-07-13

**Authors:** Ying Li, Lu Meng, Yinling Ma, Yajing Li, Xiaoqing Xing, Caihui Guo, Zhanjun Dong

**Affiliations:** 1National Clinical Drug Monitoring Center, Department of Pharmacy, Hebei Province General Center, Shijiazhuang 050051, China; lyyaoda@126.com (Y.L.); maling-shz@163.com (Y.M.); 17800317273@163.com (Y.L.); zkymw2013@163.com (X.X.); guocaihui01@163.com (C.G.); 2Hebei Provincial Key Laboratory of Basic Medicine for Diabetes, Shijiazhuang 050057, China; M18203213683@163.com; 3Shijiazhuang Technology Innovation Center of Precision Medicine for Diabetes, Shijiazhuang 050057, China

**Keywords:** UPLC-MS/MS, osimertinib, aumolertinib, furmonertinib, therapeutic drug monitoring

## Abstract

The third-generation epidermal growth factor receptor tyrosine kinase inhibitors (EGFR-TKIs), osimertinib, aumolertinib, and furmonertinib represent a new treatment option for patients with EGFR p.Thr790 Met (T790 M)-mutated non-small cell lung cancer (NSCLC). Currently, there are no studies reporting the simultaneous quantification of these three drugs. A simple ultra-performance liquid chromatography–tandem mass spectrometry (UPLC-MS/MS) method was developed and validated for the simultaneous quantitative determination of osimertinib, aumolertinib, and furmonertinib concentrations in human plasma, and it was applied for therapeutic drug monitoring (TDM). Plasma samples were processed using the protein precipitation method (acetonitrile). A positive ion monitoring mode was used for detecting analytes. D_3_-Sorafenib was utilized as the internal standard (IS), and the mobile phases were acetonitrile (containing 0.1% formic acid) and water with gradient elution on an XSelect HSS XP column (2.1 mm × 100.0 mm, 2.5 µm, Waters, Milford, MA, USA) at a flow rate of 0.5 mL·min^−1^. The method’s selectivity, precision (coefficient of variation of intra-day and inter-day ≤ 6.1%), accuracy (95.8–105.2%), matrix effect (92.3–106.0%), extraction recovery, and stability results were acceptable according to the guidelines. The linear ranges were 5–500 ng·mL^−1^, 2–500 ng·mL^−1^, and 0.5–200 ng·mL^−1^ for osimertinib, aumolertinib, and furmonertinib, respectively. The results show that the method was sensitive, reliable, and simple and that it could be successfully applied to simultaneously determine the osimertinib, aumolertinib, and furmonertinib blood concentrations in patients. These findings support using the method for TDM, potentially reducing the incidence of dosing blindness and adverse effects due to empirical dosing and inter-patient differences.

## 1. Introduction

Lung cancer, which is classified broadly as non-small cell lung cancer (NSCLC) or small cell lung cancer (SCLC), is the primary cause of cancer death worldwide and remains a major challenge [1]. NSCLC is the most common type of lung cancer, accounting for 85% of cases [2]. Therapeutic options for NSCLC have improved with the discovery of driver mutations over the past decade. Currently, molecular-targeted therapy plays a vital role in NSCLC therapies. Epidermal growth factor receptor (EGFR) mutations are the most common driver mutations in patients with advanced NSCLC, especially in Asians [3]. Molecular targeting using epidermal growth factor receptor tyrosine kinase inhibitors (EGFR-TKIs) is the first-line treatment for EGFR-mutated NSCLC [4]. However, most patients treated with first- or second-generation TKIs ultimately progress due to the emergence of the EGFR p.Thr790 Met (T790 M) point mutation. Therefore, third-generation EGFR-TKIs, including osimertinib, aumolertinib, and furmonertinib, were developed to overcome this frequently acquired mutation [5].

Osimertinib (Figure 1), is a standard first-line treatment option for patients with advanced EGFR T790 M-mutated NSCLC, effectively extending survival and improving patients’ life quality [6,7]. Subsequently, aumolertinib and furmonertinib (Figure 1) were approved in China for treating advanced EGFR T790 M-mutated NSCLC. Aumolertinib can effectively control metastatic brain lesions, with a confirmed survival benefit for patients with brain metastases [8,9,10]. As the third third-generation EGFR-TKI to be marketed in China, furmonertinib’s unique molecular structure gives it the clinical advantages of “dual activity, high selectivity, strong tumor shrinkage, and good safety,” as well as the ability to penetrate the blood–brain barrier. Furthermore, furmonertinib is gradually showing great potential for treating NSCLC [11,12,13]. However, adverse events (AEs) and EGFR-TKI resistance can emerge, leading to dose reductions or treatment discontinuation [4,14,15]. Plasma drug concentrations are closely associated with drug efficacy and side effects. A correlation between the plasma concentration of some TKIs and the occurrence of drug-related AEs and resistance to therapy was described by Yu et al. [16]. Plasma concentrations of osimertinib, aumolertinib, and furmonertinib can be affected by various factors such as pathophysiology, genetic polymorphisms, patient adherence to therapy, and interacting medications [17,18,19,20], leading to large inter-patient variability in efficacy and AEs. Therefore, it is necessary to monitor disease response and plasma drug concentrations to improve patient outcomes [21].

Therapeutic drug monitoring (TDM) is an important technique for formulating dosing regimens for drugs with strong toxic effects, and great individual differences measured by drug concentrations in biological samples can identify inter-patient differences and improve the therapeutic effects of drugs while reducing AEs [22,23,24]. Several studies have also shown that TDM can improve the therapeutic efficacy of TKIs, supporting rational clinical use [25,26,27,28]. The ultra-performance liquid chromatography–tandem mass spectrometry (UPLC-MS/MS) method is a standard analytical tool for TDM and is widely used in clinical practice for precise treatment with targeted oral drugs. Although several LC-MS/MS methods have been developed for osimertinib quantification in human plasma, few methods [29,30] have been designed for determining the blood levels of aumolertinib and furmonertinib. Furthermore, there are currently no available methods for the simultaneous quantification of osimertinib, aumolertinib, and furmonertinib in human plasma.

We developed a rapid, sensitive, and efficient UPLC-MS/MS method to simultaneously determine osimertinib, aumolertinib, and furmonertinib concentrations in human plasma. A protein precipitation method was used for sample pretreatment, and gradient elution was used to separate the analytes, which effectively reduced the measurement time and improved the detection efficiency for the target drugs. A quantitative analysis of patient plasma samples was performed to provide a basis for rational clinical drug use.

## 2. Results and Discussion

### 2.1. Method Development and Optimization

A UPLC-MS/MS method was developed for the simultaneous quantification of osimertinib, aumolertinib, and furmonertinib concentrations. The MS/MS parameters, mobile phase, and gradient elution steps used in this study were optimized to meet the requirements for the simultaneous detection of three drugs in human plasma. The runtime was 4.1 min under gradient elution, where acetonitrile was used as the organic phase. Formic acid (0.1%) was added to the organic phase to obtain distinct symmetric peaks and minimal background noise, and the final choice was acetonitrile (containing 0.1% formic acid) and water. The optimization of the mass spectrometry conditions revealed a higher response for the analytes in the positive ion mode. [M + H]^+^ was used as the parent ion for analytes because it had the best response. The target ion transitions obtained after the screening of osimertinib, aumolertinib, furmonertinib, and IS were as follows: 500.1→72.2, 526.1→72.2, 569.1→72.1, and 468.2→255.4 (Figure 2). Other MS conditions, including the declustering potential (DP), collision energy (CE), and ion source temperature (TEM), were also optimized.

Isotope-labeled internal standards are most commonly used in tandem mass spectrometry to eliminate errors due to matrix interference and differential ionization properties of the analytes. In previous studies [29,30], the IS used to detect aumolertinib and furmonertinib was a deuterated internal standard. However, deuterated aumolertinib and furmonertinib were not readily available, and d_3_-sorafenib was selected as the internal standard (IS) in this study. No significant matrix effects were observed at the retention times of the analytes. Compared to liquid–liquid extraction and solid-phase extraction, the protein precipitation method has the advantages of simplicity, speed, low cost, and reduced environmental pollution and is more suitable for TDM. Therefore, acetonitrile was selected as the protein precipitant in this study.

### 2.2. Analytical Method Validation

#### 2.2.1. Selectivity

The retention times for osimertinib, aumolertinib, furmonertinib, and IS were 1.29 min, 1.72 min, 1.80 min, and 2.34 min, respectively. At the retention times of the analytes and the IS, endogenous substances in the plasma did not interfere with the detection of each analyte, demonstrating the good selectivity and specificity of the method (Figure 3).

#### 2.2.2. Calibration Curve and LLOQ

The concentration ranges tested for osimertinib, aumolertinib, and furmonertinib were as follows: 5–500 ng·mL−1, 2–500 ng·mL−1, and 0.5–200 ng·mL^−1^, respectively. The corresponding calibration curve equations for the three analytes were: *y* = 0.0454 *x* − 0.00374 (r = 0.9995), *y* = 0.0935 *x* − 0.00211 (r = 0.9978), and *y* = 0.0868 *x* + 0.0129 (r = 0.9994), respectively. Each calibration curve showed good linearity over its respective concentration range. The lower limit of quantitation (LLOQ) concentrations for osimertinib, aumolertinib, and furmonertinib were 5 ng·mL^−1^, 2 ng·mL^−1^, and 0.5 ng·mL^−1^, respectively. Six consecutive LLOQ samples for each analyte were analyzed with the required precision and accuracy.

#### 2.2.3. Precision and Accuracy

The intra-day precision, inter-day precision, and accuracy of the quality control (QC) samples and LLOQ samples for all analytes met the requirements, and the results are within acceptable limits, as shown in Table 1.

#### 2.2.4. Matrix Effect and Extraction Recovery

The validation results for the matrix effect and extraction recovery for each compound are shown in Table 2. The results indicate that the endogenous substances did not interfere with the analyte detection.

#### 2.2.5. Stability

The stability of each analyte was examined under various storage conditions using QC plasma samples for osimertinib, aumolertinib, and furmonertinib. The results indicate the acceptable stability of the QC samples for all analytes under the storage conditions tested (Table 3).

#### 2.2.6. Carry-Over

The carry-over effect was assessed by analyzing a blank matrix sample after an upper limit of quantification (ULOQ) sample had been injected. At the retention time of the analyte, the peak areas of interfering peaks in the blank matrix sample were less than 20% of that of the LLOQ sample. Furthermore, there were no significant interfering peaks at the retention time of the IS, indicating that the high concentration sample had no carry-over effect on the determination of the low concentration sample.

### 2.3. Clinical Application

The validated method was successfully used in our laboratory to measure the plasma drug concentration at steady states of osimertinib (n = 10) and aumolertinib (n = 2) in patients with NSCLC (Table 4 and Figure 4).

Consistent with previous publications [28,31], the minimum drug concentrations (C_min_) of osimertinib at a steady state were highly variable (6.19–380 ng/mL), and the mean C_min_ was 139.98 ng/mL at the steady state. Compared to the methods previously reported, our newly developed method was more simple and convenient.

There are limited data on the C_min_ of aumolertinib and furmonertinib in patients with NSCLC. Phase I clinical trial data show that steady-state Cmin is 193 ng/mL after the administration of 110 mg of aumolertinib, and 29.1 ng/mL after the administration of 80 mg of furmonertinib [9,11]. In our study, plasma concentrations at a steady state for two patients (two samples) treated with 110 mg aumolertinib once daily were analyzed. Plasma samples from patients treated with furmonertinib were not available. The mean C_min_ of aumolertinib was 155.5 ng/mL, which is consistent with the published clinical trial data.

Brown et al. [32] did not identify a relationship between osimertinib exposure and efficacy, but a correlation between exposure and safety endpoints was observed. Due to the small number of clinical plasma samples, the relationships between the plasma concentrations of the three drugs and their efficacy and side effects were not established in our study. The collection of clinical samples from patients treated with osimertinib, aumolertinib, and furmonertinib is ongoing. There are some active metabolites of osimertinib, aumolertinib, and furmonertinib present in plasma. However, due to their low concentration [33] and difficulty in obtaining reference standards, they were not analyzed in this study, which is also a limitation of our study. The method developed in this study could be used to monitor drug plasma concentrations and the associated disease response to achieve appropriate therapeutic strategies.

## 3. Materials and Methods

### 3.1. Chemicals and Reagents

Standard reference samples of osimertinib (lot: ZC-49638, purity: 99.9%), aumolertinib (lot: ZZS-21-X043-M2, purity: 99.8%), furmonertinib (lot: ZC-47228, purity: 99.9%) were purchased from Shanghai Zhenzhun Biotechnology Co., Ltd. (Shanghai, China). IS, d_3_-Sorafenib (lot: ZZS-20-X261-A1, purity: 99.9%), was also purchased from Shanghai Zhenzhun Biotechnology Co., Ltd. (Shanghai, China). HPLC-grade acetonitrile and formic acid were obtained from Fisher Scientific (Pittsburgh, PA, USA). Ultrapure water was acquired from Wahaha Group Co., Ltd. (Hangzhou, China).

### 3.2. Chromatographic and Mass Spectrometric Determination Conditions

Chromatographic separation was performed using an ultra-high performance liquid chromatography system (LC-30 A, Shimadzu, Japan). Chromatographic separation was achieved by gradient elution on a C18 analytical column (XSelect HSS *XP*, 2.1 mm × 100 mm, 2.5 µm, Waters, Milford, MA, USA) at 40 °C. The mobile phase consisted of water (A) and acetonitrile (containing 0.1% formic acid, B). The elution procedure was carried out in the following order: 0–0.5 min, 30% B; 0.5–1.5 min, 30%→90% B; 1.5–3.5 min, 90% B; 3.5–3.6 min, 90%→30% B; 3.6–4.1 min, 30% B. The flow rate was 0.5 mL·min^−1^, and the injection volume was 8 μL.

Mass spectrometry was performed using an AB Sciex Triple Quad 4500 tandem triple quadrupole mass spectrometer equipped with an electrospray ionization source (ESI) interface. The positive ion mode with multi-reaction detection was used, and the multiple reaction monitoring transitions of the analytes were as follows: *m/z* 500.1→ 72.2(quantifier transition) and 500.1→455.3 (qualifier ion transition) for osimertinib, *m/z* 526.1→72.2 (quantifier transition) and 526.1→453.2 (qualifier ion transition) for aumolertinib, *m/z* 569.1→72.1 (quantifier transition) and 569.1→441.2 (qualifier ion transition) for furmonertinib, *m/z* 468.2→255.4 (quantifier transition) and 468.2→273.1 (qualifier ion transition) for IS (Figure 2). Other parameters were as follows: DP, 140 V (osimertinib, aumolertinib, IS), 120 V (furmonertinib); ion spray voltage, 5500 V; CE, 80 V (osimertinib, aumolertinib, and furmonertinib) and 45 V (IS), respectively; gas1, 60 psi; gas2, 50 psi; curtain gas (CUR), 25 psi; TEM, 500 °C.

### 3.3. Preparation of Stock and Working Solutions

The osimertinib, aumolertinib, and furmonertinib standards were precisely weighed and dissolved in dimethyl sulfoxide solution (DMSO) to make standard stock solutions with final concentrations of 1 mg·mL^−1^. The appropriate volume of the standard stock solution was diluted with 50% acetonitrile into working solutions with concentrations of 50, 100, 500, 1000, 2000, 3000, and 5000 ng·mL^−1^ (osimertinib), 20, 50, 100, 500, 1000, 2000, 3000, and 5000 ng·mL^−1^ (aumolertinib), and 5, 10, 50, 200, 400, 800, 1000, and 2000 ng·mL^−1^ (furmonertinib). The same method was used to prepare QC working solutions with concentrations of 150, 1500, and 3750 ng·mL^−1^ (osimertinib), 60, 1500, and 3750 ng·mL^−1^ (aumolertinib), and 15, 500, and 1500 ng·mL^−1^ (furmonertinib). The IS was also dissolved in DMSO to obtain a stock solution at a concentration of 1 mg·mL^−1^. The IS stock solution was diluted with 50% acetonitrile to obtain a working solution at 500 ng·mL^−1^. All stock solutions and working solutions were stored at −20 °C.

### 3.4. Plasma Sample Preparation

Plasma samples were processed using the protein precipitation method. Ten microliters of IS (200 ng·mL^−1^) were added to 100 μL of plasma sample (including calibration curve or QC samples (10 μL of calibration standard solution or QC working solution were added to 90 μL of blank plasma)). Then, 300 μL of acetonitrile were added, vortexed for 1 min, and centrifuged at 12,000 rpm for 10 min. The supernatant (100 µL) was added to 300 μL of 50% acetonitrile, vortexed and mixed, and finally transferred to an autosampler vial for sample analysis.

### 3.5. Method Validation

The method was comprehensively validated for selectivity, calibration curve linearity, the LLOQ, precision and accuracy, matrix effect, extraction recovery, stability, and carry-over according to the US FDA [34] and Chinese Pharmacopoeia (2020) Guidelines for the Validation of Bioanalytical Methods.

#### 3.5.1. Selectivity

The method’s selectivity was determined by assessing the interference of other components in plasma. Plasma samples containing 2 ng·mL^−1^ aumolertinib were prepared by sequentially adding the aumolertinib working solution (20 ng·mL^−1^) to blank plasma from different sources (n = 6). The samples were processed as described in Section 3.4. Plasma samples containing osimertinib and furmonertinib were also prepared using this procedure, and the plasma was then analyzed. In the absence of interference, the peak area of the analyte in the blank plasma should be less than 20% of the LLOQ and 5% of the IS within the retention time.

#### 3.5.2. Calibration Curve and LLOQ

Using the concentration of the analyte as the horizontal coordinate (*x*), the peak area ratio of the analyte to the IS as the vertical coordinate (*y*), and 1/*x^2^* as the weighting factor, a weighted least squares method was used to obtain the regression equation and generate the calibration curve. The linear ranges of the standard curves for osimertinib, aumolertinib, and furmonertinib were 5–500 ng·mL^−1^, 2–500 ng·mL^−1^, and 0.5–200 ng·mL^−1^, respectively. The difference between the back-calculated and the nominal concentrations for each standard in the calibration curve had an acceptable range (<15%). For LLOQ, the difference should be less than 20%.

#### 3.5.3. Precision and Accuracy

The precision and accuracy were assessed by measuring low, medium, and high concentrations of QC (LQC, MQC, HQC) samples and LLOQ samples. Intra-day and inter-day precision and accuracy were calculated by repeating the measurements six times a day for three consecutive days for samples at each concentration level. Precision was expressed by calculating the relative standard deviation (RSD) of the samples from six parallel measurements. Accuracy was expressed by the relative error (RE) of the samples. The RSD and RE were within ±15% for the QC samples and within ±20% for the LLOQ samples.

#### 3.5.4. Matrix Effect and Extraction Recovery

The matrix effect for the analyte was determined by comparing the analyte peak area in the blank plasma samples from six different donors at LQC and HQC levels (n = 6 for each level) with the analyte peak area in pure solution. The extraction recovery was assessed by comparing the analyte peak area in extracted plasma samples at LQC, MQC, and HQC levels (n = 6 for each level), with the analyte peak area of a blank plasma extract spiked at the same level.

#### 3.5.5. Stability

The stability of the samples under different storage or handling conditions was assessed by analyzing the QC samples at different levels (LQC, MQC, HQC; n = 6). The short-term stability was assessed after placing the plasma samples at room temperature for 8 h. Quality control samples were stored at −20 °C for 7 d to assess the long-term stability. We also examined the stability of the samples stored in a refrigerator at 4 °C for 24 h. The stability of the processed samples was assessed by storing the samples in an autosampler at 4 °C for 24 h. The samples stored at −20 °C were thawed at room temperature, and three freeze–thaw cycles were performed to investigate the freeze–thaw stability of the samples.

#### 3.5.6. Carry-Over

The carry-over of this method was investigated by sequentially injecting the LLOQ and the ULOQ samples, followed by a blank biological matrix sample. The area of interfering peaks at the retention time of the analyte in the blank plasma sample should be less than 20% of the LLOQ and 5% of the IS peak areas.

### 3.6. Clinical Samples Analysis

All the experimental procedures were approved by the Ethics Committee of Hebei General Hospital (No. 2022094).

Plasma samples from patients who had been administered osimertinib, aumolertinib, or furmonertinib for at least two weeks and were within half an hour before re-administration were analyzed to demonstrate the applicability of the assay. The clinical blood samples were residual blood samples obtained from patients for other routine clinical measurements at Hebei General Hospital. Blood was collected in EDTA-containing anticoagulation tubes, and the supernatant was separated by centrifugation at 3000 rpm for 10 min at 4 °C. Then, the supernatant was transferred to 1.5 mL EP tubes and stored at −20 °C until analysis.

## 4. Conclusions

We have developed a simple, rapid, and sensitive UPLC/MS/MS method to simultaneously determine osimertinib, aumolertinib, and furmonertinib concentrations in human plasma. The validated method may also be a useful tool for the TDM of the third-generation EGFR-TKIs for NSCLC patients in clinical practice.

## Figures and Tables

**Figure 1 molecules-27-04474-f001:**
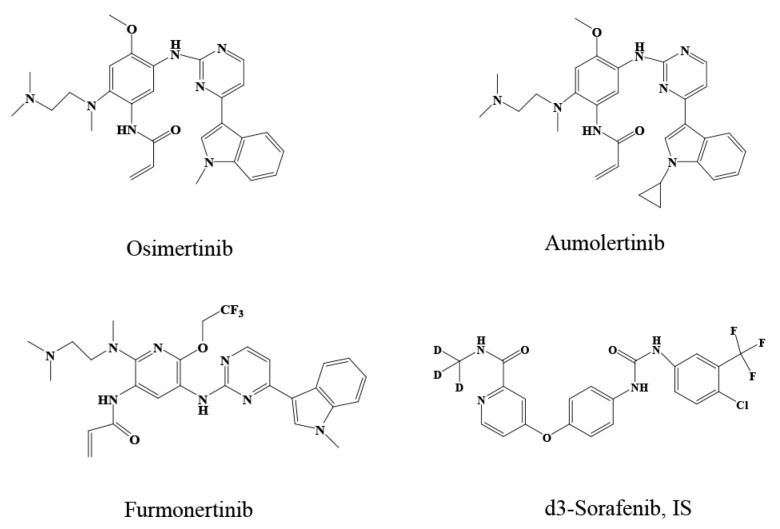
Chemical structures of osimertinib, aumolertinib furmonertinib, and IS.

**Figure 2 molecules-27-04474-f002:**
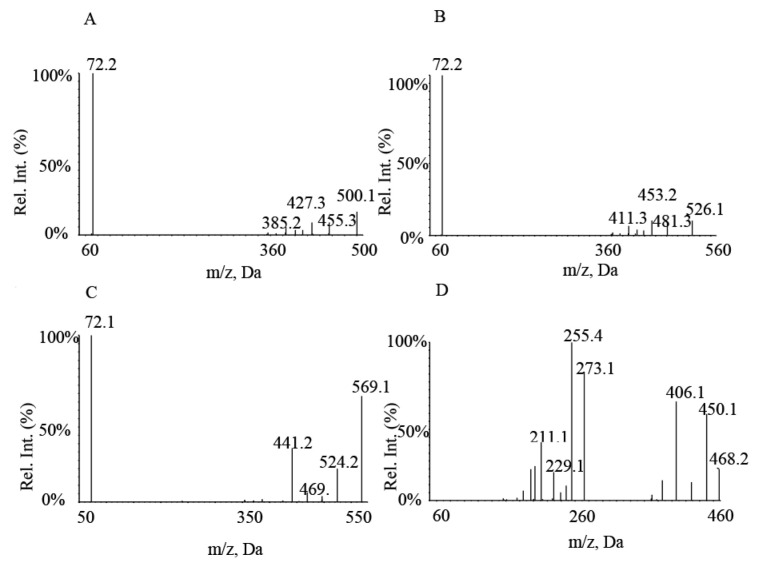
Product ion spectra of (**A**) osimertinib, (**B**) aumolertinib, (**C**) furmonertinib, and (**D**) IS.

**Figure 3 molecules-27-04474-f003:**
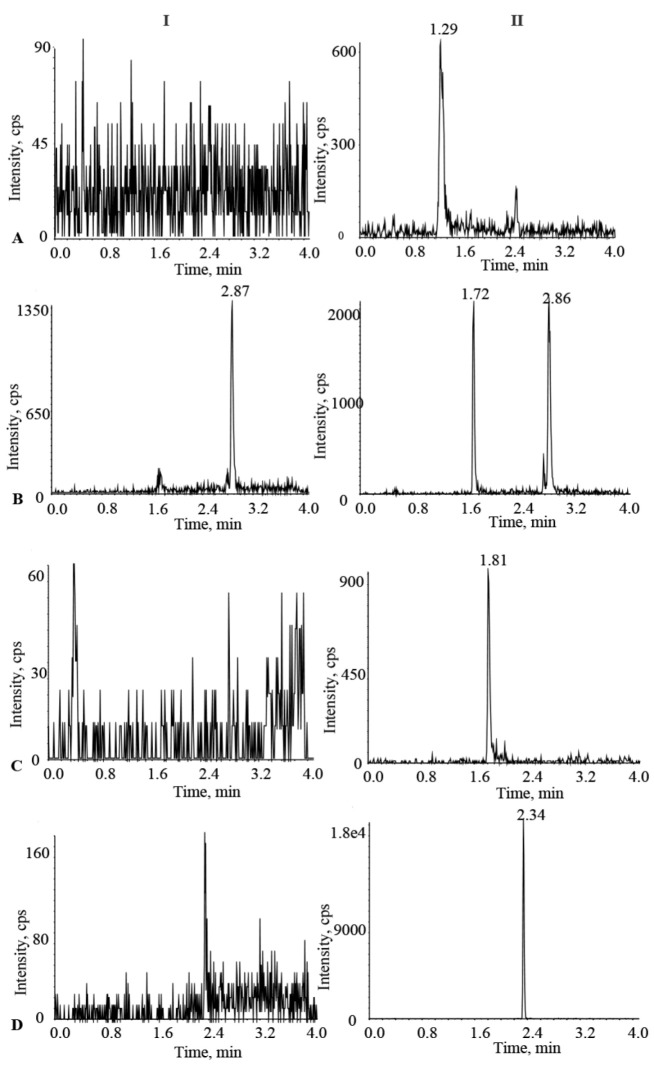
The chromatogram of analytes and IS in blank plasma (**I**). The chromatogram of analytes and IS at LLOQ concentration levels (**II**); (**A**) osimertinib; (**B**) aumolertinib; (**C**) furmonertinib; (**D**) IS.

**Figure 4 molecules-27-04474-f004:**
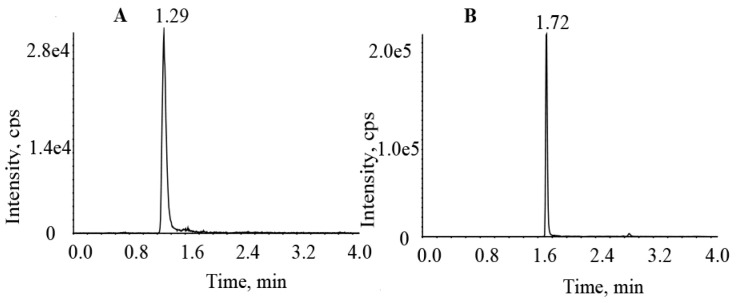
Chromatogram of plasma samples from patients treated with (**A**) osimertinib and (**B**) aumolertinib.

**Table 1 molecules-27-04474-t001:** Precision and accuracy data of osimertinib, aumolertinib, and furmonertinib in human plasma (n = 6).

Analyte	Spiked Conc. (ng/mL)	Intra-Day (n = 6)			Inter-Day (n = 18)		
		Mean ± SD (ng/mL)	Precision (RSD%)	Accuracy (RE%)	Mean ± SD (ng/mL)	Precision (RSD%)	Accuracy (RE%)
Osimertinib	5.0	4.84 ± 0.25	5.3	96.9	4.78 ± 0.05	1.0	95.8
	15.0	14.47 ± 0.42	2.9	96.4	14.49 ± 0.24	1.7	96.6
	150.0	145.0 ± 4.20	2.9	96.6	151.72 ± 5.92	3.9	101.1
	375.0	369.2 ± 20.58	5.6	98.3	382.39 ± 13.87	3.6	101.9
Aumolertinib	2.0	2.14 ± 0.06	3.0	95.8	2.06 ± 0.13	6.1	103.3
	6.0	6.31 ± 0.30	4.8	105.1	6.23 ± 0.34	5.4	103.8
	150.0	157.8 ± 6.01	3.8	105.2	156.2 ± 5.18	3.3	104.1
	375.0	376.3 ± 17.60	4.7	100.4	380.8 ± 13.70	3.6	101.6
Furmonertinib	0.5	0.52 ± 0.03	5.6	104.1	0.51 ± 0.03	5.6	102.6
	1.5	1.47 ± 0.08	5.2	97.8	1.42 ± 0.06	4.2	94.7
	50.0	52.1 ± 2.10	4.0	104.2	49.9 ± 1.97	4.0	99.9
	150.0	154.3 ± 4.80	3.1	102.9	151.8 ± 5.16	3.4	101.2

**Table 2 molecules-27-04474-t002:** Matrix effect and extraction recovery of analytes in human plasma (n = 6).

Analyte	Spiked Conc. (ng/mL)	Extraction Recovery		Matrix Effect	
		Mean ± SD (%)	RSD (%)	Mean ± SD (%)	RSD (%)
Osimertinib	15.0	98.78 ± 4.87	4.9	106.00 ± 4.89	4.6
	150.0	102.81 ± 4.95	4.8	/	/
	375.0	101.67 ± 5.23	5.2	97.30 ± 4.93	5.1
	6.0	103.05 ± 9.14	8.9	103.33 ± 8.33	8.1
Aumolertinib	150.0	102.41 ± 5.87	5.7	/	/
	375.0	102.32 ± 5.68	5.6	96.28 ± 1.88	2.0
	1.5	100.49 ± 5.62	5.6	92.34 ± 0.06	5.28
Furmonertinib	50.0	99.71 ± 4.50	4.5	/	/
	150.0	99.56 ± 3.07	3.1	102.95 ± 0.05	5.25

**Table 3 molecules-27-04474-t003:** Stability of analytes in human plasma under various storage conditions (mean ± SD, n = 6, %).

Analyte	Spiked Conc. (ng/mL)	Room Temperature for 8 h in Human Plasma	−20 °C for 7 d in Human Plasma	4 °C for 24 h in a Refrigerator	Placed in an Automatic Sampler at 4 °C for 24 h	3 Freeze-Thaw Cycles, −20 °C to Room Temperature
Osimertinib	15.0	96.42 ± 4.61	98.43 ± 4.68	98.37 ± 4.06	96.05 ± 3.54	100.18 ± 5.57
	150.0	102.53 ± 2.96	101.53 ± 2.68	105.17 ± 4.67	100.97 ± 2.32	103.73 ± 2.75
	375.0	100.25 ± 4.38	101.23 ± 4.53	104.60 ± 4.51	102.03 ± 5.73	106.22 ± 4.16
Aumolertinib	6.0	98.05 ± 4.69	105.50 ± 3.51	101.11 ± 7.26	101.17 ± 5.34	101.38 ± 3.78
	150.0	100.01 ± 3.36	105.00 ± 2.19	104.48 ± 4.58	97.70 ± 1.77	104.00 ± 3.69
	375.0	101.27 ± 5.06	103.18 ± 2.53	104.53 ± 7.20	98.48 ± 7.49	101.58 ± 3.90
Furmonertinib	1.5	96.60 ± 2.27	96.20 ± 3.74	100.00 ± 6.45	92.78 ± 5.16	101.93 ± 4.88
	50.0	104.20 ± 5.10	99.98 ± 1.56	93.95 ± 2.18	105.58 ± 4.25	99.27 ± 4.40
	150.0	99.18 ± 2.83	97.35 ± 3.19	93.55 ± 1.44	102.17 ± 1.60	97.82 ± 2.22

**Table 4 molecules-27-04474-t004:** Steady-state trough concentrations of analytes in patient plasma.

Analyte	Dosage	Mean Plasma Concentration (ng/mL)	Concentration Range (ng/mL)
Osimertinib	80 mg, qd	139.98 (n = 10)	6.19–380
Aumolertinib	110 mg, qd	155.5 (n = 2)	131–180

## Data Availability

Not applicable.

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
