# Peer review of "Determination of Osimertinib, Aumolertinib, and Furmonertinib in Human Plasma for Therapeutic Drug Monitoring by UPLC-MS/MS"

_molecules, 2022, doi:10.3390/molecules27144474_

Round 1
Reviewer 1 Report
The manuscript is well organized and shows novelty for some drugs' bioanalysis, I do agree publishing it after the authors consider the following remarks as a minor revision:
1. It is suggested to add the structure of IS to allow a structural comparison with the analytes.
2. FDA guidelines for bioanalysis validation should be added to the references list.
3. In page 3, under method development; product ion spectra are recommended to be added to confirm the selection of these specific daughter ions for the 3 drugs.
4. Is there any explanation for the appearance of a peak at 2.87 min in blank plasma in figure 2 (I) B?
5. In page 6, it was mentioned that the obtained results are consistent with previous publications [28, 31], referring to reference [28] similar work was done for osimertinib, authors can compare the reported methods that are common with their suggested work.
6. the clinical application was carried out on only two patients that can't guarantee considering all factors of variation e.g. age, weight, gender..etc
Author Response
Zhanjun Dong
National Clinical Drug Monitoring Center, Department of Pharmacy, Hebei Province General Center
No.348 Heping west Road, Shijiazhuang, 050051, Hebei, China
Tel: +86-0311-8598-8604
E-mail: [email protected]
July 10, 2022
Dear Editors and Reviewers:
Thank you for your letter and for the reviewers’comments concerning our manuscript entitled “Determination of osimertinib, aumolertinib, and furmonertinib in human plasma for therapeutic drug monitoring by UPLC-MS/MS”(ID: molecules-1807706). Those comments are all valuable and very helpful for revising and improving our paper, as well as the important guiding significance to our researches. We have studied comments carefully and have made correction which we hope meet with approval. Revised portion are marked in red in the paper.
The main corrections in the paper and the responds to the reviewer’s comments are shown below.
Thank you for your patience and time.
General Reply:
Many thanks to the Reviewer for encouraging our work and for giving useful comments for clarifying, improving and correcting some materials in the paper. Now we have carefully revised the paper according to your comments, as explained below:
1. It is suggested to add the structure of IS to allow a structural comparison with the analytes.
Relpy: Thank you very much for your helpful suggestions on our manuscript. We have revised the manuscript appropriately. We have added the structure diagram of IS, as shown in Figure 1.
2. FDA guidelines for bioanalysis validation should be added to the references list.
Relpy: Special thanks to you for your valuable’s comments and suggestion. Those comments are very helpful for revising and improving our paper. We have made revision according to the Reviewer’s comments. (See in section 3.5)
3. In page 3, under method development; product ion spectra are recommended to be added to confirm the selection of these specific daughter ions for the 3 drugs.
Relpy: Thank you very much for this valuable comment. These comments are very meaningful for revising and improving our manuscript. We have added the product ion spectra of four drugs in a timely manner in response to the reviewers' comments. (See in Figure 2)
4. Is there any explanation for the appearance of a peak at 2.87 min in blank plasma in figure 2 (I) B?
Relpy: Thanks a lot for your kind comments. These comments have a good guiding significance for our research. The peak at 2.87 min, which could be a plasma endogenous substance, but as it was not at the same retention time as the analyte, we did not find any significant interference from endogenous substances at the retention time of the analyte, which proves the good specificity of the method, and therefore we did not pay particular attention to this component.
5. In page 6, it was mentioned that the obtained results are consistent with previous publications [28, 31], referring to reference [28] similar work was done for osimertinib, authors can compare the reported methods that are common with their suggested work.
Relpy: Thank you very much for this valuable comment. Those comments are all valuable and very helpful for revising and improving our paper. In fact, the chromatographic conditions and sample pre-treatment were optimized in order to obtain satisfactory peak shapes, sensitivity and selectivity. Compared with previously reported detection methods [28], our mobile phase composition and sample handling method are more convenient. In addition, our extraction method also appears to have higher recoveries. Simple mobile phase composition and sample processing steps greatly shorten the measurement time of the entire sample. The retention time of osimertinib was shortened from 2.70min to 1.29min. Compared with the methods previously reported, our newly developed method was more simple and convenient. (See in section 2.3)
6. The clinical application was carried out on only two patients that can't guarantee considering all factors of variation e.g. age, weight, gender..etc
Relpy: Special thanks to you for your valuable’s comments and suggestion. Due to the short time on the market of aumolertinib, the number of patients taking this drug is still relatively small. Although the work of collecting a sufficient number of samples is still in progress, the sample size we have collected so far is only two cases, and the plasma concentrations of these two patients do not fully reflect the inter-individual variation. Based on your valuable suggestions, we have adjusted the manuscript appropriately. We have removed the inter-individual variation item. (See in Table 4)
Reviewer 2 Report
Li et al. developed a UPLC-MS/MS method for the quantification of three 3rd generation TKIs osimertinib, aumolertinib, and furmonertinib, which have recently been approved for the treatment of EGFR-T790M-mutated NSCLC, in human plasma.
This is a well planned and conducted study. Validation of the method has been done according to international guidelines. However, there are some concerns that need to be addressed prior publication. A major issue is the inclusion of active metabolites in the method. The authors need to proof that the inclusion of these metabolites would not bring additional value to the clinical application of the method.
General comments:
All three drugs are metabolized by CYP3A4 to active N-demethylated metabolites. These metabolites may also contribute to efficacy and ADEs. The inclusion of these metabolites in the quantitative analysis could be of help in explaining variability in clinical outcome. At lest the authors need to give references for their statement that metabolite concentrations in patient plasma are very low and therefore negligible.
Detailed comments:
Line 26: Could you please specify acceptable (e.g. according to FDA / ChP 2020 guidelines)
Line 52: Aumolertininib
Line 68f: Resolution of the figure could be improved. In addition, please add the structure of the internal standard.
Line 100ff: some guidelines recommend a second ion transition as qualifier in addition to the quantifier.
Line 100: ... ion pairs -> ion transitions
Line 102: … mass conditions -> MS conditions
Line 106: To my knowledge, 13CD3-osimertinib is available at Toronto Research Chemicals. This would have been an alternative to sorafenib-d3. The different structure of sorafenib results in a higher retention time compared to the analytes. This might be a problem during the analysis of clinical samples from patients as different coeluting matrix compounds might have different influences on IS and analytes, respectively.
Lines 264ff: In this context, matrix effects investigated with blank plasma obtained from different patients to account for disease specific changes are indicated. It is no clear whether blank plasma form different donors were used here.
By the way, the stable isotope labelling does not give additional value (unless sorafenib is present in the samples, too).
Line 142: Has this been tested in 6 blank plasma samples from 6 different healthy volunteers or NSCLC patients?
Line 182: Could you please provide a reference for low plasma concentrations of active metabolites as this is an important issue and might be a limitation of the present method.
Line 198: please add supplier of the column
Line 198ff: I guess, the gradient program given here is wrong. It should start with a high percentage of water followed by increasing percentage of acetonitrile. -> mobile phase A: water; mobile phase B: 0.1% formic acid in acetonitrile. Is there any rational why formic acid was added to acetonitrile but not to water?
Line 204: please revise this sentence.
Line 241: as described in Section 2.4 -> … section 3.4
Line 260: Please clarify the number of replicates used for the determination of inter-day accuracy and precision: 6 or 18?
Author Response
Zhanjun Dong
National Clinical Drug Monitoring Center, Department of Pharmacy, Hebei Province General Center
No.348 Heping west Road, Shijiazhuang, 050051, Hebei, China
Tel: +86-0311-8598-8604
E-mail: [email protected]
July 10, 2022
Dear Editors and Reviewers:
Thank you for your letter and for the reviewers’comments concerning our manuscript entitled “Determination of osimertinib, aumolertinib, and furmonertinib in human plasma for therapeutic drug monitoring by UPLC-MS/MS”(ID: molecules-1807706). Those comments are all valuable and very helpful for revising and improving our paper, as well as the important guiding significance to our researches. We have studied comments carefully and have made correction which we hope meet with approval. Revised portion are marked in red in the paper.
The main corrections in the paper and the responds to the reviewer’s comments are shown below.
Thank you for your patience and time.
General Reply:
Many thanks to the Reviewer for encouraging our work and for giving useful comments for clarifying, improving and correcting some materials in the paper. Now we have carefully revised the paper according to your comments, as explained below:
General comments: All three drugs are metabolized by CYP3A4 to active N-demethylated metabolites. These metabolites may also contribute to efficacy and ADEs. The inclusion of these metabolites in the quantitative analysis could be of help in explaining variability in clinical outcome. At lest the authors need to give references for their statement that metabolite concentrations in patient plasma are very low and therefore negligible.
Relpy: Special thanks to you for your valuable’s comments and suggestion. These comments are very meaningful for revising and improving our manuscript. We have made revision according to the Reviewer’s comments appropriately (See in manuscript). We have sorted out the reasons for not choosing metabolites, as follows:
- Determination of active metabolites may be helpful for the therapeutic effect of drugs. For example, some psychotropic drugs in therapeutic drug monitoring require monitoring of active metabolites [1]. The ratio of active metabolites to the parent can be used to preliminarily determine the source of drug concentration variation. However, for TKIs, this definite proportional relationship has not yet been established. The currently recommended monitoring of TKI tumor drugs is still monitoring the parent drug. For example, sorafenib also has active metabolites, but it is recommended to monitor sorafenib [2].
- It is primarily the parent drug that exerts therapeutic effects and causes adverse drug reactions in vivo. At present, the correlation study between the efficacy and adverse reactions is limited to the parent drug, and the correlation between the exposure of active metabolites and the efficacy and adverse reactions has not been determined. Therefore, monitoring the parent drug is more clinically relevant than the active metabolite.
- The concentration of active metabolites is usually low and not easily detectable, such as osimertinib, active metabolites only account for about 10% [3]. As reference standards for metabolites are not readily available and their cost is high, especially for newly launched drugs such as aumolertinib and furmonertinib.
This method was developed based on a study of TDM and needs to be considered practical for everyday clinical application. We did not include active metabolites in the establishment of the method, but we have included it as a limitation and put it into the article. (See in section 2.3)
Detailed comments:
Line 26: Could you please specify acceptable (e.g. according to FDA / ChP 2020 guidelines)
Relpy: Thank you very much for this valuable comment. We have revised the manuscript appropriately. (See in abstract)
Line 52: Aumolertininib
Relpy: Thanks a lot for your kind comments. We have made careful revisions in the article. (See in introduction)
Line 68f: Resolution of the figure could be improved. In addition, please add the structure of the internal standard.
Relpy: Thank you very much for your helpful suggestions on our manuscript. We have increased the resolution of the figure appropriately. We have also added the structure diagram of IS, as shown in Figure 1.
Line 100ff: some guidelines recommend a second ion transition as qualifier in addition to the quantifier.
Relpy: Special thanks to you for your valuable’s comments and suggestion. In MRM analyses, we monitored both quantifier and qualifier ion for each drug. In section 3.2, the qualifier transitions have been added.
Line 100: ... ion pairs -> ion transitions
Relpy: Thank you very much for your helpful suggestions on our manuscript. We have made careful revisions in our paper. (See in section 2.1)
Line 102: … mass conditions -> MS conditions
Relpy: Thanks a lot for your kind suggestions. We have revised the manuscript in time. (See in section 2.1)
Line 106: To my knowledge, 13CD3-osimertinib is available at Toronto Research Chemicals. This would have been an alternative to sorafenib-d3. The different structure of sorafenib results in a higher retention time compared to the analytes. This might be a problem during the analysis of clinical samples from patients as different coeluting matrix compounds might have different influences on IS and analytes, respectively.
Relpy: Special thanks to you for your valuable’s comments and suggestion. These comments have a good guiding significance for our research. For osimertinib, 13CD3-osimertinib is indeed a good internal standard choice. For aumolertinib and furmonertinib, isotopes were difficult to obtain due to the late availability of the drug, however, after methodological validation, d3-sorafenib was found to be feasible as an internal standard for osimertinib, aumolertinib and furmonertinib. In the application process of the later method, we also keep an eye on the differences in matrix effects on internal standards and analytes and use isotopic internal standards when they are available. To my knowledge, d3-sorafenib is not clinically co-administered with these three drugs and its use is more cost-effective than stable isotopic internal standards.
Lines 264ff: In this context, matrix effects investigated with blank plasma obtained from different patients to account for disease specific changes are indicated. It is no clear whether blank plasma form different donors were used here.
Relpy: Thank you very much for this valuable comment. Those comments are all valuable and very helpful for revising and improving our paper. We apologize for the imprecise presentation in our manuscript. The matrix effect was assessed using six different donors of blank plasma, which we have modified in the article. (See in section 3.5.4)
By the way, the stable isotope labelling does not give additional value (unless sorafenib is present in the samples, too).
Relpy: Thanks a lot for your kind suggestions. Neither sorafenib nor d3-sorafenib will be used simultaneously with the three drugs mentioned above. In China, d3-sorafenib is also marketed as another drug, called donafenib. At the same time, after methodological verification, it was proved that d3-sorafenib can be applied, so we chose it as the internal standard.
Line 142: Has this been tested in 6 blank plasma samples from 6 different healthy volunteers or NSCLC patients?
Relpy: Special thanks to you for your valuable’s comment. The matrix effect was tested in 6 blank plasma samples from 6 different NSCLC patients.
Line 182: Could you please provide a reference for low plasma concentrations of active metabolites as this is an important issue and might be a limitation of the present method.
Relpy: Thank you very much for this valuable suggestion. This is indeed a limitation of our study, which has been described in the text. (See in section 2.3)
Line 198: please add supplier of the column
Relpy: Thanks a lot for the reviewer’s suggestion. We have added relevant information. (See in section 3.2)
Line 198ff: I guess, the gradient program given here is wrong. It should start with a high percentage of water followed by increasing percentage of acetonitrile. -> mobile phase A: water; mobile phase B: 0.1% formic acid in acetonitrile. Is there any rational why formic acid was added to acetonitrile but not to water?
Relpy: Thanks a lot for your kind suggestions. We apologize for our mistakes. We have revised the manuscript in time. (See in section 3.2)
When we explored the mobile phase conditions in the early stage, we tried to use the ammonium acetate-acetonitrile and formic acid water-acetonitrile used in the previous research conditions as the mobile phase combination. However, we found that after adding ammonium acetate or formic acid to the aqueous phase, the baseline noise of aumolertinib was high, but when formic acid was added to acetonitrile, the baseline noise decreased, and the response values and peak shapes of the three analytes were not affected, and finally water-acetonitrile (0.1% formic acid) was used as the mobile phase.
Line 204: please revise this sentence.
Relpy: Thank you very much for this valuable comment. We have revised the manuscript appropriately.
Line 241: as described in Section 2.4 -> … section 3.4
Relpy: Thanks a lot for your kind suggestions. We have revised the manuscript in time.
Line 260: Please clarify the number of replicates used for the determination of inter-day accuracy and precision: 6 or 18?
Relpy: Special thanks to you for your valuable’s comments and suggestion. These comments are very meaningful for revising our manuscript. We have made timely changes in the corresponding tables. (See in Table 1)
References
[1] C. Hiemke. Consensus Guideline Based Therapeutic Drug Monitoring (TDM) in Psychiatry and Neurology, Current drug delivery 13(3) (2016) 353-61.
[2] H. Yu, N. Steeghs, C.M. Nijenhuis, J.H. Schellens, J.H. Beijnen, A.D. Huitema. Practical guidelines for therapeutic drug monitoring of anticancer tyrosine kinase inhibitors: focus on the pharmacokinetic targets, Clinical pharmacokinetics 53(4) (2014) 305-25.
[3] P.A. Dickinson, M.V. Cantarini, J. Collier, P. Frewer, S. Martin, K. Pickup, P. Ballard. Metabolic Disposition of Osimertinib in Rats, Dogs, and Humans: Insights into a Drug Designed to Bind Covalently to a Cysteine Residue of Epidermal Growth Factor Receptor, Drug Metab Dispos 44(8) (2016) 1201-12.